# Foliar Transpiration Inhibitor Reduces Cd Accumulation in Rice Grain: The Potential Effect of the Endophytic Bacterial Community

**DOI:** 10.3390/toxics13090755

**Published:** 2025-09-05

**Authors:** Ge Lei, Huijuan Song, Ziwen Gan, Yunchou Yang, Anwei Chen

**Affiliations:** 1Department of Environment & Ecology, Hunan Agricultural University, Changsha 410128, China; lgdaniel2022@163.com (G.L.); 13807462800@163.com (Z.G.); anweihuanjing@126.com (A.C.); 2Department of Civil and Environmental Engineering, The Hong Kong Polytechnic University, Hong Kong, China; 3Agricultural and Rural Affairs Bureau of Dao County Hunan Province, Yongzhou 425300, China; dxnyzh123@163.com

**Keywords:** paddy field, Cd, endophytic bacteria, translocation factor, adsorptive site

## Abstract

Excess Cd in soils can be accumulated in rice, presenting a serious human health risk. The effect of foliar transpiration inhibitors (TIs) on the Cd content and the endophytic bacterial community in rice plants was unclear. We evaluated the key part of the rice plant to control the Cd translocation and the profile of the endophytic bacterium structure after spraying with foliar reagents; some possible typical endophytes were induced by the TIs to inhibit the Cd translocation in the rice plant. The rice plants in three sites with different available Cd content were sprayed with foliar TIs. We assessed the Cd, N, P, K and water-soluble saccharide (WSS) in different parts of the rice plant and the endophytic bacteria community in the stem. Foliar application of TIs reduced Cd translocation factor (*TF*_Cd_) by ~20% from the root to the grain compared with that of CK. The TI can increase the adsorptive site concentration of stem nodes from 5.10 to 6.83 mmol/g. The diversity of the endophytic bacteria community was enhanced after application of TI, and the Shannon index increased from 3.29 to 3.92. The endophytic bacterial community induced by TI showed higher potentiality on the biofilm and stress-tolerant and metal-transport functions than that of CK, respectively. The relative abundances of *Burkholderiaceae* and *Bacterium*_g_*Anaeromyxobacter* were significantly negatively correlated (*p* < 0.05), with *TF*_Cd_ and positively correlated (*p* < 0.05), with water-solution saccharide content, simultaneously. The TI enhanced the endophytic diversity and amount. A high abundance of special endophytic bacteria induced by TI might decrease the *TF*_Cd_.

## 1. Introduction

China faces huge challenges in soil pollution, according to the first national soil pollution survey [1]. Heavy metal is the main source of soil pollution, accounting for 82.4%, where cadmium (Cd) is the primary, accounting for 7%. It is highly mobile in soils and can be transferred through the food chain, causing damage to human health and ecosystems [2,3]. Cd tends to accumulate in rice plants, the main food crop in China [4,5,6]. Excessive Cd consumption can cause “itai-itai disease”, as exemplified by the eating of Cd-contaminated rice in Japan in the 1960s. Therefore, how to effectively reduce the content of Cd in rice grain has become an urgent matter for the Chinese.

Foliar spray reagents are becoming a more and more important and effective technology to reduce the Cd in rice grain. The foliar reagents include six kinds [1,7]: (1) the ion antagonistic effect, such as Mn, Zn, Fe, and Ca; (2) the deposition effect, such as Si and Se; (3) the complexation effect, such as Glutathione and Cystathionine; (4) the transpiration inhibition effect, such as humic acid and salicylic acid; (5) the antioxidation effect, such as anthocyanin; and (6) the ion-channel inhibition effect, such as La and Gd. The ion antagonistic effect and transpiration inhibition were investigated relatively deeply in reducing the Cd accumulation in grain and then were applied widely in practical rice production.

The uptake and translocation of Cd in the rice plant are affected significantly by transpiration because the Cd ion is easily migrated in solution. Transpiration inhibitors (TIs) control plant transpiration and are often used to improve transplant survival. Their main components are humic acid and a lot of nutritional elements such as nitrogen, phosphorus, and potassium. The TI decreases the stomatal conductance of the leaf at high temperature periods and then reduces the Cd translocation through the low-transpiration effect. It was reported that high temperature increased Cd accumulation in rice grains through enhancing the transpiration effect [8,9]. And high-Cd-accumulation crop species are usually positively correlated with a strong transpiration effect. Therefore, the foliar application of TIs can reduce Cd uptake and accumulation in rice grain in the flowering and filling stages.

Much effort has been made to understand the mechanisms of reducing Cd accumulation in crop plants after the application of foliar reagents; however, little research has focused on the effects on the endophytic bacterial community structure. In fact, foliar reagents (containing plant hormones and other organic and inorganic compounds) can significantly influence both the activity and composition of endophytic microbial communities. These foliar reagents alter nutritional or metabolic profiles, which may in turn affect endophytic microbial activity, ultimately leading to the suppression of heavy metal translocation from soil to grains [10]. Studies have shown that foliar application of SiO_2_; nanoparticles can alter the soil microbial community composition in the rhizosphere of Pakchoi (*Brassica chinensis* L.) grown in contaminated mine soil [11]. It is implied that the foliar reagent can act not only on the endophytes directly but also on the bacteria indirectly. Furthermore, the endophyte can adsorb the metal and restrict its mobilization. It is reported that the fusaric acid from a mangrove endophyte can capture the metal as Cd-, Cu-complexes [12].

The mechanisms by which endophytic microbes affect Cd translocation in rice plants primarily include adsorption, biosequestration, and the regulation of gene expression. The strong metal-binding ability of the endophytic bacterium induced the low metal translocation efficiency in the plant [13]. Endophytic *Falciphora oryzae* can sequester Cd in its vacuoles and chlamydospores, thereby enhancing Cd tolerance in rice and decreasing the translocation efficiency to shoots [14]. Alternatively, endophytic microbes can regulate the expression of some important channel proteins to adjust the uptake of essential nutrients along with mediating the uptake and translocation of heavy metals as well [15]. After the *Stenotrophomonas maltophilia* R5-5 was inoculated in rice plants, the expression of OsNramp5 and OsHMA2 was down-regulated in the root [16], and the Cd accumulation in rice grain was reduced. Typically, the Cd translocation process influenced by endophytes in the plant tends to involve more than one mechanism. However, the effect of foliar spray reagents on the endophytic community structure and the relation with Cd translocation factor (*TF*) in rice plants is unclear.

In this study, foliar reagents of TI were sprayed on rice at the flowering and filling stages, and the Cd concentration, *TF* and the profile of endophytic bacterial community within the plants were measured. The purpose of this experiment is to explore (1) the key part of rice plant to control the Cd translocation; (2) the profile of the endophytic bacterium structure after spraying with foliar reagents; and (3) some possible typical endophytes inhibiting the Cd translation in rice plant. Our investigation might provide a new viewpoint to understand the reducing of Cd accumulation in the rice grain after the foliar reagents, which could have a potential impact on the safety of rice production.

## 2. Materials and Methods

### 2.1. Experiment Site and Rice Cultivar

Field trials were conducted in Changfeng Village, Liuyang City, Hunan province, China. The soil Cd concentration at the site exceeded the national soil environmental quality standard (0.3 mg kg^−1^). The basic physical–chemical properties in different sites are shown in Table 1. A randomized block design with three replicates was employed. The plot area was 20.0 m^2^, with 5.0 m length and 4.0 m width.

The seed of rice (*Oryza sativa* L. cv. Zhongzao 39) was supplied by the China Rice Research Institute. Seeds were planted in March and seedlings were transplanted in late April; its whole growth period is 112 days. Field managements were as same as those used in local production. Weeds were controlled by chemical herbicide treatment.

### 2.2. The Components of TI and Application

The TI with humic acid purchased from the Sichuan Guoguang Agricultural Chemical Co., Ltd. (Jianyang, China) The humic acid content was 40 g/L and the nutrition element of N, P_2_O_5_, K_2_O was 40 g/L, 80 g/L, 150 g/L, respectively. This TI solution was diluted 1000 times with water to foliar application.

### 2.3. Experimental Design

Beginning in mid-July 2019, the TI solution was first sprayed onto rice plants at the heading stage and then applied once per week until the grouting stage. In total, the plants received three foliar applications during the entire growth period. These foliar reagents were sprayed onto the leaves with a handheld sprayer. Control plants (CKs) were sprayed with water, and every 20 m^2^ plot was sprayed with 2.0 L foliar reagents/water. At the maturity stage, three plants with roots and topsoil (0–20 cm) were collected by a shovel from the center of each plot. The roots were rinsed with tap water to remove soil. Subsequently, the samples were transported to the laboratory and washed three times with deionized water. The harvested plants were then separated into grains, stems, and roots. These grains and stems were dried for 6h at 70 °C, and the roots were stored in a constant temperature refrigerator at −70 °C.

### 2.4. Determination and Analysis

#### 2.4.1. The Metal Concentration

The concentrations of Cd in plant tissues were measured with ICP (ICPMA 8300, Perkin-Elmer, Shelton, CT, USA). A 2.0 g aliquot of powdered sample was weighed into a 50 mL dry glass digestion tube. Then, 10 mL of concentrated nitric acid was added, and the mixture was left to stand overnight. The digestion tube was placed into the digestion furnace to accelerate digestion. The temperature of the digestion solution was raised to 70 °C in 15 min, kept for 30 min, and then heated up to 90 °C in 30 min; finally, it was maintained at 120 °C for 2 h until the funnel digestion became clear. After cooling to room temperature, the digest was diluted with ultrapure water to 40 mL and filtered through Whatman filter paper, and then measured by inductively coupled plasma–optical emission spectrometry (ICP-OES, Optima 5300, Perkin-Elmer, Shelton, CT, USA).

For the available Cd of soil, the soil sample was extracted with 0.01 M CaCl_2_ solution using a soil/solution ratio of 1:5 (m:m), shaken for 2 h, and then filtered. The extract was analyzed with ICP-OES.

#### 2.4.2. The Water-Soluble Saccharide (WSS) Content

Amounts of 0.5 g fresh stem node samples were ground in a pre-cooled mortar and subjected to extraction with a small amount of quartz sand and 5% trichloroacetic acid (TCA) solution. The sample was then centrifuged at 4000× *g* for 10 min at 4 °C, and the supernatant was collected for the measurement of the WSS content using the Bradford method [17].

#### 2.4.3. The Nutrition for Endophytic Microorganism in the Rice Stem

We weighed 0.5000 g of ground rice stems and added 25 mL of a 2% acetic acid solution at 0.5 mol/L. This was shaken for extraction for 30 min at 25 °C and 200 rpm. Then, it was centrifuged at 10,000× *g* for 15 min, the supernatant was filtered through a 0.45 μm membrane, and the resulting filtrate was used for measurement. The soluble nitrogen (SN), soluble phosphorus (SP), and soluble potassium (SK) levels were determined using potassium persulfate oxidation UV spectrophotometry, molybdenum antimony colorimetric method and inductively coupled plasma optical emission spectroscopy (ICP-OES), respectively [18].

#### 2.4.4. Adsorptive Site Concentration Measured with Potentiometric Titration

Potentiometric titrations were performed with the Metrohm 905 Titrando system (905 Titrando, Metrohm, Herisau, Switzerland), coupled with a GK2401C combination glass electrode. The glass electrode was calibrated using buffers at pH of 4.01, 7.00 and 10.00 before each titration. The concentration of the standard HCl and NaOH solutions were tested before the titration.

The fresh stem nodes were washed with distilled water and ethanol three times and then freeze-drying to constant weight. The 0.50 g dried stem nodes were ground to powders using liquid N_2_ in a mortar. The powders were placed in a beaker with 70 mL of ultrapure water (18 MΩ·cm, 25 °C) with 0.001 M NaCl, under a N_2_ atmosphere at 25 °C and then titrated with 0.1 M NaOH and 0.5 M HCl solutions. A known amount of HCl was added at the beginning of the experiment to lower the pH to approximately 2.5. The composites were equilibrated for 40 min and then titrated to pH 10 with NaOH. The background value was titrated with the same treatment using deionized water. Subsequently, a non-electrostatic model was used to fit the potentiometric titration data. The p*K*_a_ (type of functional group), total adsorption sites, and site concentration were calculated using FITEQL4.

### 2.5. Bacterial Richness and Diversity

The stem sections from three different locations at each level of contamination were combined and thoroughly mixed. The total microbial DNA was then extracted from each sample using the Soil Master DNA Extraction kit (Epicentre Biotechnologies, Madison, WI, USA) following the manufacturer’s instructions. The V4-V5 region of the 16S ribosomal RNA (rRNA) gene was amplified using the universal primers 515F (GTGCCAGCMGCCGCGG) and 907R (CCGTCAATTCMTTTRAGTTT) (i.e., 515907).

The oligonucleotide sequence barcode was fused to the forward primer. The PCR reaction mixture (20 μL) consisted of 4 μL of 5 FastPfu reaction buffer (TransGen Biotech, Beijing, China), 2 μL of a dNTP mixture (2.5 mM), 0.4 μL of each primer (5 mol/L), 0.4 μL of FastPfu polymerase, 10 ng of template DNA, and H_2_O to reach the final volume. PCR thermal cycling was performed with an initial denaturation at 95 °C for 5 min, followed by 25 cycles of denaturation at 95 °C for 30 s, annealing at 55 °C for 30 s, and extension at 72 °C for 30 s, with a final extension period of 5 min at 72 °C. Following PCR amplification, the products were examined on a 2% (*w*/*v*) agarose gel and purified using the Maxiprep DNA Gel Extraction Kit (Axygen Biosciences, Union City, CA, USA). The purified amplicons were quantified using QuantiFluor-ST (Promega, Fitchburg, WI, USA) and then sequenced in paired-end fashion on an Illumina MiSeq platform at Majorbio Bio-Pharm Technology Co., Ltd. (Shanghai, China), following standard protocols.

The resulting sequences were clustered into operational taxonomic units (OTUs) using a 97% sequence similarity threshold. Taxonomic ranks were assigned to representative OTU sequences using the naive Bayesian Classifier (v.2.2) of the Ribosomal Database Project (RDP). Bacterial community diversity and composition were subsequently analyzed based on the OTU assignments and taxonomic classifications.

### 2.6. Bioinformatics and Statistical Analysis

Raw pyrosequencing data were de-multiplexed and quality-filtered using the Trimmomatic tool in the way Lohse et al. [19] described. Overlapping reads were then merged into single long reads with the FLASH software tool (https://www.cbcb.umd.edu/, accessed on 27 August 2025) [20]. Qualified sequences were then clustered into OTUs at a 97% similarity cutoff using Usearch v7.1 (http://qiime.org/, accessed on 15 November 2019). The phylogenetic affiliation of each 16S rDNA sequence was analyzed with the RDP Classifier v2.2 (http://sourceforge.net/projects/rdp-classifier/, accessed on 20 November 2019), using a confidence threshold of 0.7 and the reference database Silva (Release 115, http://www.arb-silva.de, accessed on 23 November 2019). The heatmap figures were produced using package ‘gplots’ in the R (v3.1.1) software (http://www.Rproject.org/, accessed on 29 November 2019).

Gene Abundance Quantification. The abundance of each gene in the non-redundant catalog within each sample was quantified by mapping the high-quality, host-filtered reads back to the catalog using Bowtie2 (v2.4.4). The number of reads mapped to each gene was counted. The abundance of a given gene was calculated and normalized as reads per kilobase per million mapped reads (RPKM) to account for gene length and sequencing depth variations between samples. The formula is:RPKM = (Number of reads mapped to the gene × 10^9^)/(Total number of mapped reads × Gene length in bases)

This normalized value represents the relative abundance of a specific gene in the microbial community.

Functional Annotation. The non-redundant protein sequences were aligned against the Kyoto Encyclopedia of Genes and Genomes (KEGG) database (Release 106.0) using Diamond (v2.0.15) with a blastp search (*e*-value ≤ 10^−5^). Genes were assigned to KEGG Orthology (KO) entries based on the best hit.

Identification and Abundance Profiling of Metal-Transport-Related Enzymes. KO entries associated with metal transport functions were identified by querying the KEGG BRITE hierarchies and pathway maps (e.g., ko02010 for ABC transporters, ko01100 for metabolic pathways) for relevant terms.

The relative abundance of a specific metal-transport-related enzyme in a sample was represented by the sum of the RPKM values of all non-redundant genes annotated to the corresponding KO entry. This value serves as a proxy for the functional potential of that enzyme within the endophytic bacterial community.

The overall functional profile for metal transport was constructed by aggregating the abundances of all KO entries belonging to this category.

The translocation factor (*TF*) of Cd from A part to B part of rice was defined as *TF*_A–B_. These *TF* values were estimated as follows:$${TF}_{root\text{-}rice}=\frac{{Cd}_{rice}}{{Cd}_{root}}$$$${TF}_{stem\text{-}rice}=\frac{{Cd}_{rice}}{{Cd}_{stem\;node}}$$$${TF}_{root\text{-}stem}=\frac{{Cd}_{stem\;node}}{{Cd}_{root}}$$
where *Cd_rice_*, *Cd_root_* and *Cd_stem node_* are the concentrations (mg/kg, dry weight) of Cd in rice grain, roots, and stem nodes, respectively.

### 2.7. Statistical Analyses

The statistical analysis of the obtained results using the SPSS 19.0. Every site included two treatments: foliar TI spray treatment and control. Every treatment included three samples as three replicates. Due to the small sample size, although we conducted a normality test, its effectiveness was insufficient. Considering that this indicator is usually considered to be approximately normally distributed, we used the Student’s *t*-test to examine statistical significance of differences in different groups. Mean ± SE (standard error) was calculated using three replicates for each treatment.

The correlation between relative abundance of endophytic bacterial species and physical parameters was determined using Spearman analysis. Prior to correlation analysis, the normality of the data distribution was assessed using the Shapiro–Wilk test.

## 3. Result

### 3.1. Effects of TI on Cd Concentration in Different Organs of Rice Plants

The Cd contents in the grains, stem nodes, and roots of rice from paddy fields at different sites are shown in Figure 1a–c. The Cd content followed the order root > stem > grain. In rice grains, TI controlled the concentration of Cd in rice below the national safety standard of 0.2 mg/kg at site B and site C (Figure 1a). The available Cd concentration in sites A, B and C were 0.21 mg/kg, 0.15 mg/kg and 0.12 mg/kg, respectively. In the stem nodes, the Cd concentration under the three treatments all decreased with decreases in the available Cd concentration in the soil (Figure 1b). The TI treatment resulted in relatively higher Cd concentrations in the stem nodes compared to the CK. These results indicate that the TI treatment promoted the accumulation of Cd in the stem nodes. The stem nodes of rice represent an important pool for Cd accumulation, which may regulate the translocation of Cd to the leaves and grains.

In the roots, Cd concentration reduced with the decrease in the available Cd concentration in the soil, in the TI treatments (Figure 1c). Generally, foliar application was performed with TI-sprayed aboveground parts, regulating the Cd concentration in aboveground stem and rice grain, and with no significant change in the underground root. The Cd concentration in roots was 22–30 mg/Kg, which was not a significant variation between different sites. This indicated that the Cd accumulation in roots was affected mainly by the cultivar, not the environmental Cd stress. However, the *TF_root-rice_* decreased significantly with the decrease in the available Cd concentration in soil. Furthermore, the *TF_root-rice_* with TI treatment was markedly reduced in site B and site C compared with site A (Figure 1d). These results revealed that the *TF_root-rice_* was affected by not only the foliar reagents, but also the soil properties.

To further understand the *TF* variation in different treatments, the histograms of *TF* in different translocation pathways of rice plants following different treatments are shown in Figure 2. The *TF_stem-rice_* of 0.01–0.06 was relatively lower than *TF_root-stem_* of 0.2–0.7. The TI can induce a higher *TF_root-stem_* and a lower *TF_stem-rice_*. It was revealed that the TI induced a higher Cd content in stem node than that of CK.

### 3.2. The WSS Content in Stem Nodes

The WSS content of stem nodes increased with decreasing the available Cd concentration in the soil, as shown in Figure 3a. Furthermore, the WSS increased at the same site with TI application than that of CK. The highest content of WSS was 0.40%, corresponding to the sample with TI treatment at site C. The relation between Cd *TF_root-rice_* and the WSS content is revealed in Figure 3b. The negatively linear relation was fitted with an *R* of 0.84. This indicated that the high WSS content corresponded to a low Cd translocation efficiency. This finding is consistent with previous reports that Cd concentration and its binding to the cell wall fraction are strongly positively correlated with lactose concentration [21]. Therefore, the WSS seems to be able to promote the Cd being adsorbed or immobilized in the organs in plants.

### 3.3. The Adsorptive Site Concentration of Stem Nodes

The results of the cadmium adsorption capacity of stem nodes are shown in Table 2. The total adsorptive site concentration of TI and CK in site C were 6.83 and 5.10 mmol/g, respectively. The total adsorptive site concentration increased more in TI than that of CK, at all sites. This result corresponded to the TI inducing lower *TF_root-rice_* of Cd than that of CK, because of a high concentration of adsorptive sites for Cd in stem nodes. The functional groups present on biosurfaces are divided into three types [22], where p*K*_a_ values of 4–6, ~7, and 8–11 correspond to carboxyl group, phosphate group, and amino/hydroxyl group, respectively. The TI treatment increased the amino/hydroxyl groups, as shown in Table 2.

### 3.4. Effects of TI on Biological Species in Rice Plants

The foliar reagents reduce the Cd content in the rice grain; this is usually attributed to the Cd translocation efficiency from the root to the rice grain being depressed. Based on the above results, the bottom node is the key organ to restrict the Cd transport upward. The mechanism might be mainly related to the high Cd adsorption capacity and chemical chelation effects. The foliar reagent can change the internal environment of the rice plant, induce the variation of the endophytic bacterial community, and then reduce the Cd transport. Therefore, the endophytic bacterial community was analyzed through a high-throughput sequencing technique to explore the potential relation between the microbial species and the Cd translocation upward in the rice plant.

High-throughput sequencing technique based on the 16S rRNA revealed the significant changes in microbial species diversity. The average total OTUs number was 622 and 716 in samples with CK (A, B, C samples) and TI (ATI, BTI, CTI samples) treatment, respectively. The Shannon index of A, B, C, ATI, BTI and CTI were 3.43 ± 0.6, 3.38 ± 0.4, 3.06 ± 0.4, 3.45 ± 0.4, 3.63 ± 0.3 and 4.68 ± 0.9, respectively. The average value of the Shannon index were 3.29 and 3.92 for CK and TI, respectively. These results are shown in Figure 4a. This indicates that the foliar TI application can improve the environment for the endophytic bacterial community and increase the microbial species diversity.

The relative abundance of main microbial species at the phylum level changed after the application of foliar reagents, as shown in Figure 4b. The Proteobacteria, Actinobacteria, Bacteroidetes, Chloroflexi were the main phylum. Their relative abundance followed the order Proteobacteria > Actinobacteria > Bacteroidetes > Chloroflexi. A heatmap depicting the endophytic bacterial community at the species level is shown in Figure 4c. The species with relative abundance higher than 1% included the *bradyrhizobium* sp_g_*Afipia*, *Bacterium*_g_*Anaeromyxobacter*, *Paraburkholderia kururiensis*, *Burkholderiaceae*, *Ralstonia solanacearum*_g_*Ralstonia*, *Oryza sativa Indica Group long-grained rice* g norank f *Mitochondria.*

Uncultured *bradyrhizobium* sp_g_*Afipia* decreased by 9.2% under TI treatment. Oryza sativa Indica Group long-grained *rice*_g_norank_f *Mitochondria* in TI treatment decreased by 5.48%. In contrast, *Paraburkholderia kururiensis*, unclassified_f_*Burkholderiaceae*, *Bacterium*_g_*Anaeromyxobacter* and *Ralstonia solanacearum*_g_*Ralstonia* increased by 1.2% to 6.25% under TI treatment. The *bradyrhizobium* sp_g_*Afipia* and *Bacterium*_g_*Anaeromyxobacter* are the typical N_2_-fixation endophytic bacteria. *Paraburkholderia kururiensis* and unclassified_f_*Burkholderiaceae* are two types of plant growth-promoting bacteria. TI treatment can regulate the endophytic bacterial community by significantly increasing the relative abundance of plant growth-promoting bacteria and decreasing the abundance of some non-promoting bacteria.

### 3.5. The Main Predicted Function of Endophytic Bacterial Community in Different Treatments

Using the two main heavy-metal-related functions, including stress-tolerant and forms-biofilm of endophytic bacterial community induced by CK, TI treatments were predicted using the BugBase tool (version 1.0, JSON). The relative abundances of main endophytes associated with these two functions are shown in Figure 5a,b. For the forms-biofilm and stress-tolerant functions, the TI treatment showed higher function potential than that of the CK. The functions of forms-biofilm and stress-tolerant were related to the Cd-binding and biosequester abilities of the endophytic bacterial community in rice plant.

The contributions of endophytes to the forms-biofilm function mainly included *Afipia*, *Anaeromyxobacter*, *Pleomorphomonas*, *Pseudolabrys*, *Mesorhizobium*, etc. The highest relative abundance to contribute this function was *Afipia* in TI treatment, with a relative abundance of 12%.

The endophytes contribution to the stress-tolerant function mainly included unclassified_f__*Burkholderiaceae*, *Burkholderia*_*Caballeronia*_*Paraburkholderia*, *Ralstonia*, *Acidovorax*, unclassified_f_*Rhodocyclaceae*. The relative abundance of unclassified_f__*Burkholderiaceae* was as high as 23% in TI, as the most crucial contributor.

### 3.6. Enhanced Cd Biosequestration Potential in Endophytes

To further illustrate the potential ability of endophytes to biosequester Cd, the abundance of the metal-transport-relative enzyme in endophytes induced by different treatments is presented in Table 3. The abundance of P-type Ca^2+^/Zn^2+^/Mg^2+^/Mn^2+^/Cu^2+^ transporter and Cd^2+^-exporting ATPase for endophytes induced by TI treatment was higher than that by CK. The Cd^2+^ can be transported into endophytes’ cells through these P-type Ca^2+^/Zn^2+^/Mg^2+^/Mn^2+^/Cu^2+^ transporters [23]. Furthermore, the high abundance of Cd^2+^-exporting ATPase indicates that endophytes in the TI treatment possess an enhanced capacity for Cd detoxification. It was demonstrated that the endophytes in rice plants reduced by TI had the highest Cd-sequestering ability compared to that by CK. And this result corresponded to the endophytes with high stress-tolerant function induced by TI treatment.

### 3.7. Identification of Typical Endophytes and Their Environmental Relevance

In this study, nutrients (N, P, K), WSS and Cd, and *TF_root-grain_* in rice stem were taken into consideration to evaluate the relative contributions to endophytic bacteria at the species level. The detailed correlation between the bacterial species and the environmental parameters are shown in Figure 6. A species-level heatmap split the six parameters into two groups at the first level. One was composed of Cd, *TF_root-grain_* and K, N, while the other was composed of WSS, P.

Notably, unclassified_g__*Mycobacterium* was positively correlated with Cd (R = 0.77, 0.001 < *p* ≤ 0.01). Metagenome_g_*Pseudolabrys* was positively correlated with Cd (*R* = 0.89, 0.01 < *p* ≤ 0.05) and K (*R* = 0.83, 0.01 < *p* ≤ 0.05). *Burkholderiaceae* was negatively correlated with Cd (*R* = −0.94, 0.001 < *p* ≤ 0.01), *TF_root-gra__in_* (*R* = −0.88, 0.01 < *p* ≤ 0.05), while positively corresponded to WSS (*R* = 0.89, 0.01 < *p* ≤ 0.05). Uncultured_bacterium_g__*Anaeromyxobacter* was negatively correlated with *TF_root-gra__in_* (*R* = −0.86, 0.01 < *p* ≤ 0.05), while positively corresponded to WSS (*R* = 0.86, 0.01 < *p* ≤ 0.05).

The relationship between the WSS content and the microbial species was investigated as well. Two microbes were positively correlated with WSS (*p* ≤ 0.05), i.e., *Burkholderiace* and *Bacterium*_g_*Anaeromyxobacter*. This suggests that an increase in the relative abundance of these specific endophytic bacteria may contribute to a decrease in the *TF_root-rice_*. Figure 7 presents the detailed MaAslin analysis of the relationships between the two microbes and both *TF_root-rice_* and WSS content. The high relative abundance of *Burkholderiace* and *Bacterium*_g_*Anaeromyxobacter* was revealed in the rice plant with different treatments, respectively.

## 4. Discussion

### 4.1. The TI Enhancing the Cd Accumulation in the Stem Node

TI treatment resulted in a higher *TF_root-stem_* and a lower *TF_stem-rice_*, suggesting that Cd accumulation was enhanced in the stem nodes rather than in the rice grain. The foliar reagents reduce all the translocation efficiency of Cd in every part, in that the change of Cd accumulation in rice is related to Cd transport genes. The OsHMA3 [24], NRAMP5 [25], and OsHMA2 [26] have been found to play a role in Cd absorption/transport in rice, and the gene expression of these transporters is usually related to the application of foliar reagents [27]. Prior to the application of the foliar reagent, a substantial amount of Cd had already accumulated in the stem nodes. Subsequently, the transportation of Cd was depressed after the foliar application with TI at the heading stage of rice plant growth. Therefore, TI induced the relatively lower *TF_stem-grain_*, and the Cd was accumulated at the stem nodes.

Our results revealed that the stem nodes exhibited a higher Cd accumulation capacity after spraying with TI, due to their higher concentration of adsorptive sites compared to the CK. The adsorptive sites were mainly attributed to the cell wall, endophytes and the extracellular polymeric substances (EPS) secreted by endophytes. However, the precise relationships between specific functional groups and individual components could not be unequivocally identified in this study. Mainly, the hydroxyl group was correlated with a silicon-containing cell wall, and the phosphate and amino groups were related to the EPS [28] and endophytes [29]. Therefore, the high adsorptive site concentration might be attributed to the large amount of endophytes. Furthermore, this was supported by the observation that the concentration of WSS was also higher in TI-treated plants than in CK-treated plants.

The WSS content might be correlated with the amount of endophytes. On the measurement process, the endophytes were extracted from the rice plant as the component of WSS. Therefore, a high number of endophytes induced by foliar TI resulted in the high adsorptive site concentration in rice plant to reduce the Cd translocation upward in rice plant.

### 4.2. TI Enhancing the Endophytic Diversity and Cd Uptake Capacity

TI increased the endophytic community diversity. Foliar spraying of TI changed the community structure of the colonizing bacteria and regulated the internal ecological environment of rice plant. Tian et al. [11], spraying foliar nano SiO_2_ on pakchoi, found that the rhizosphere bacterial and fungal were increased. And these bacteria promoted the carbon and nitrogen cycle, which led to increasing root secretion and the production of heavy metal adsorption chelation, thereby improving the pollution-resistance effect. It was indicated that the foliar reagent could change the relative abundance of endophytic bacterial species, through altering the internal nutrition and ecological environment in plants.

TIs increase the forms-biofilm and the stress-tolerant functions of endophytes. The biofilms are usually composed of extracellular polysaccharide, protein, and lipid. The biofilms have been considered to have a high Cd-binding capacity, and with a large adsorptive site [30]. Therefore, an enhanced biofilm-forming capability likely contributes to a greater Cd-binding effect. Normally, the stress tolerance of endophytes to heavy metal partly reflects its detoxification effect, because the high heavy metal concentration can induce the high content of antioxidant enzymes, to alleviate the reactive oxygen and heavy metal damage [31]. The endophytic microbes with excellent Cd biosequestering capacity might correspond to their high detoxification effect; otherwise, the high Cd content would damage its physiological activity.

### 4.3. Typical Endophytes Potential Effect on Reduce Cd Translocation

Our research results indicate that among rice plants subjected to different treatments, both *Burkholderiace* and *Bacterium*_g_*Anaeromyxobacter* exhibit relatively high abundance. *Burkholderia* cepacian complex bacteria usually have high resistance to Cd, Cu, Zn, Co, Pb and Hg [32]. It is noted that the *Burkholderia* can colonize in the root system of plants [33] and compete with the root cells to sequester Cd, which reduces the Cd upward transport. Wang et al. [34] found that the *Burkholderia* sp. Y4 inoculation in rice plant would increase essential nutrient uptake and inhibit Cd accumulation in rice by preferential Cd-biosequester.

The *Bacterium*_g_*Anaeromyxobacter* can drive reductive dechlorination of chlorophenols [35]. *Anaeromyxobacter* spp. can defend against other bacteria using exoenzymes [36]. The *Bacterium*_g_*Anaeromyxobacter* induced increasing the Cd concentration and decreasing the Cu concentration in the roots. The endophytes can improve the Cd binding on the cell wall surface through the function groups such as -OH, -NH_2_, -PO_4_ and -COOH. The *Anaeromyxobacter* strain can secrete abundant EPS [35] to adsorb Cd as well. Similar research [37] showed that Cd-immobilizing’s *Serratia liquefaciens* CL-1resulted in reducing Cd translocation upward in rape plant. Therefore, the large number of special endophytes might induce high adsorption effect of rice plant to Cd, and then decrease *TF_root-rice_* of Cd.

## 5. Conclusions

The foliar spray application of TIs can efficiently reduce the Cd translocation from root to rice and decrease the Cd accumulation in the brown rice in the paddy soil with different available Cd concentration. The low Cd transport from root to rice grain was mainly restricted by the nodes because of its high adsorptive site concentration. The foliar TI enhances the endophytic bacterial diversity. The relative abundance of *Bacterium*_g_*Anaeromyxobacter* and *Burkholderiaceae* were increased by TI treatments, respectively. Furthermore, the endophytic bacterial community induced by TIs showed greater ability under forms-biofilm and stress-tolerant functions than that by CK. It was indicated that the endophytes induced by TI revealed a greater potential for Cd-binding and bio-sequestering effect, respectively. The relative abundance of *Bacterium*_g_*Anaeromyxobacter* and *Burkholderiaceae* were significantly correlated with the low Cd transport upward. Our findings demonstrated these endophytic bacteria’s diversity and relative abundance can be regulated by the foliar reagents. And some key endophytic microbes would regulate Cd translocation efficiency.

## Figures and Tables

**Figure 1 toxics-13-00755-f001:**
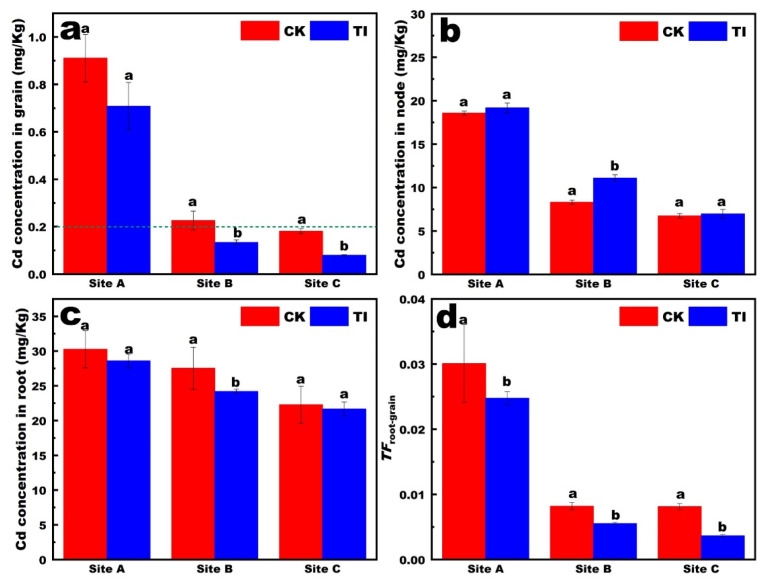
Effect of TI on Cd concentrations in rice grains (**a**); stem nodes (**b**); roots (**c**) and Cd translocation factor from root to rice grain (**d**) of rice plants grown in Cd-contaminated field with different Cd concentration (The letters a, b indicate statistically significant differences between treatments by Student’s *t*-test. Mean ± SE (standard error) was calculated using three replicates for each treatment).

**Figure 2 toxics-13-00755-f002:**
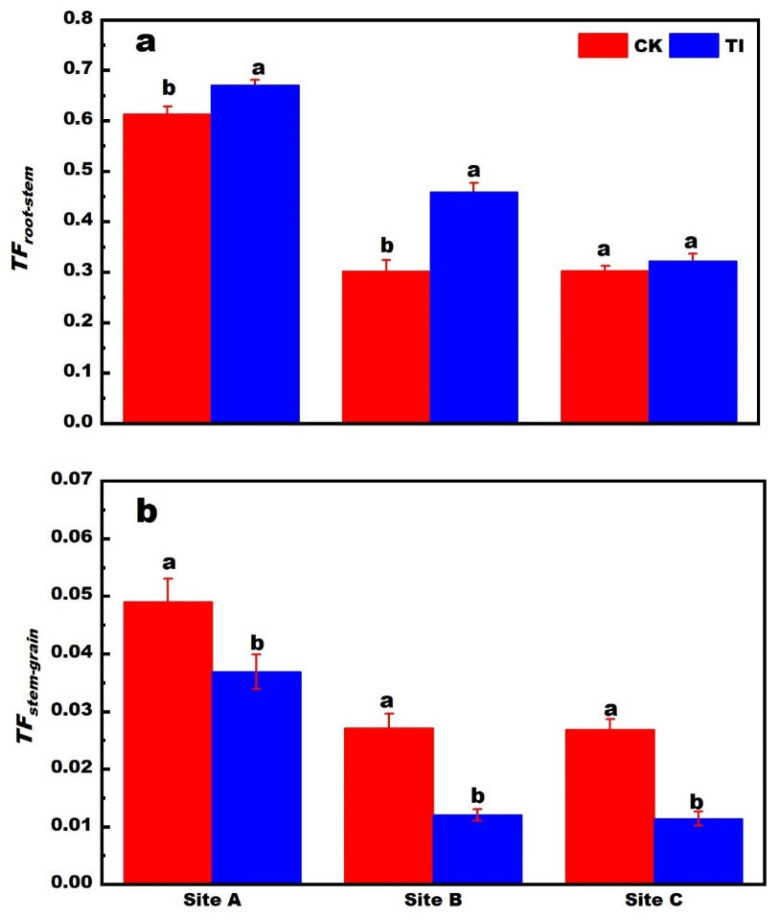
The translocation factors (TF) of root to stem nodes (**a**), and (**b**) stem node to rice grain, following the CK and T1 treatments in different sites (the letters a, b indicate statistically significant differences between treatments by Student’s *t*-test. Mean ± SE (standard error) was calculated using three replicates for each treatment).

**Figure 3 toxics-13-00755-f003:**
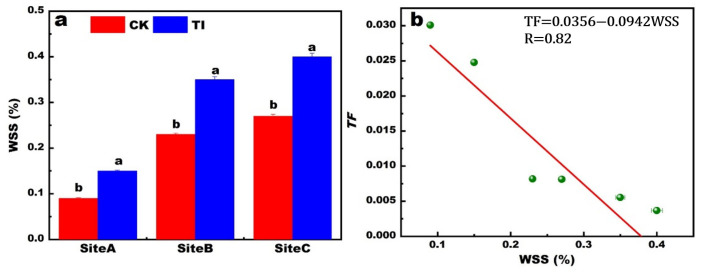
The content of water-soluble saccharide in stem nodes of rice plant (**a**). The relation between the *TF_root-grain_* and the WSS concentration (**b**). (The letters a, b indicate statistically significant differences between treatments by Student’s *t*-test. Mean ± SE (standard error) was calculated using three replicates for each treatment).

**Figure 4 toxics-13-00755-f004:**
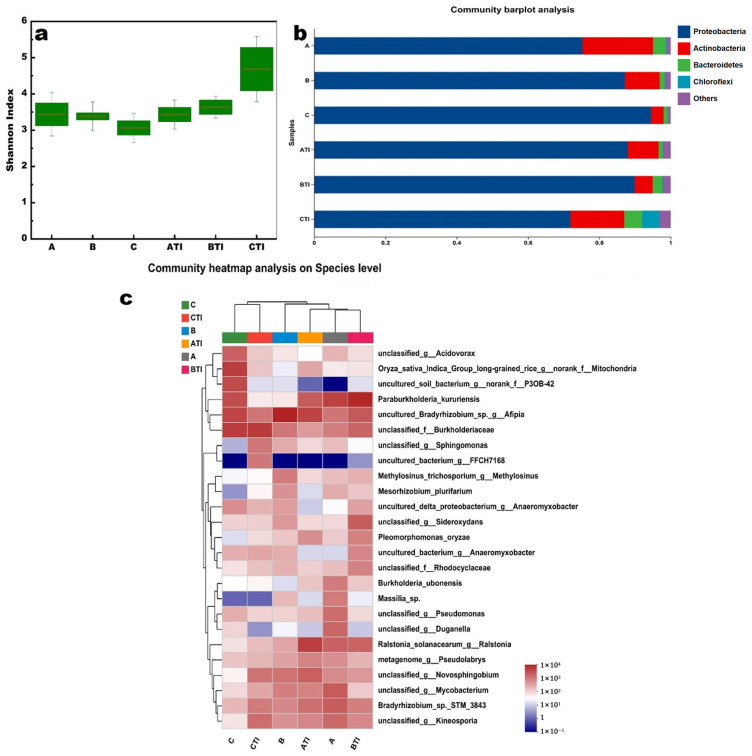
The Shannon index of different treatments with A, B, C, and ATI, BTI, CTI (**a**). The abundance of endophytic bacterial community on phylum level in the rice plant with different treatments (**b**). The community heatmap on species level of different treatments (**c**). The A, B, C was the CK in site A, B, C and the ATI, BTI, CTI was the samples in site A, B, C after treating with TI, respectively. Mean ± SE (standard error) was calculated using three replicates for each treatment.

**Figure 5 toxics-13-00755-f005:**
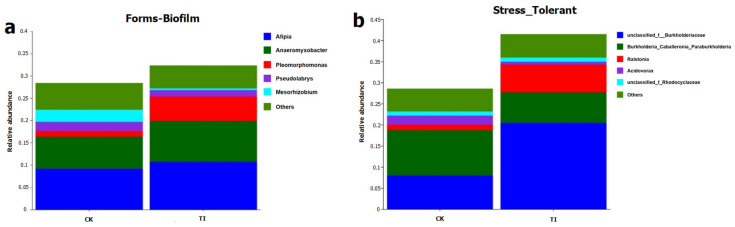
The relative abundance of main endophytic bacteria leading to forms-biofilms (**a**) and stress-tolerant (**b**) function in the rice plant with CK and TI treatments.

**Figure 6 toxics-13-00755-f006:**
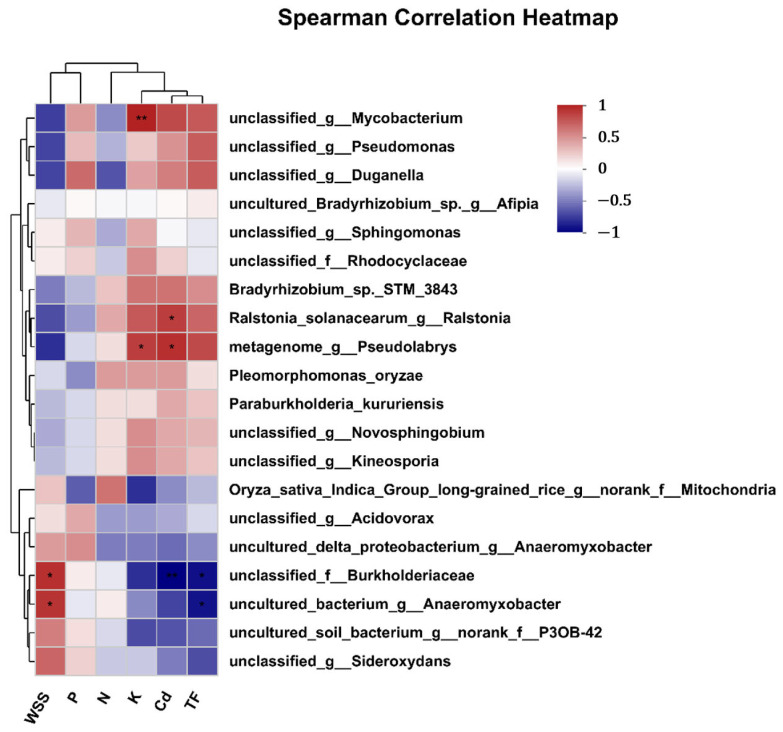
The heatmap of the correlation between relative abundance of endophytic bacterial species and physical parameters. WSS, P, N and K refer to the content of water-soluble saccharide, SP, SN and SK in the rice stem, respectively. And Cd, TF refer to the content of Cd in grain and *TF_root-gra__in_*, respectively (* 0.01 < *p* ≤ 0.05, ** 0.001 < *p* ≤ 0.01).

**Figure 7 toxics-13-00755-f007:**
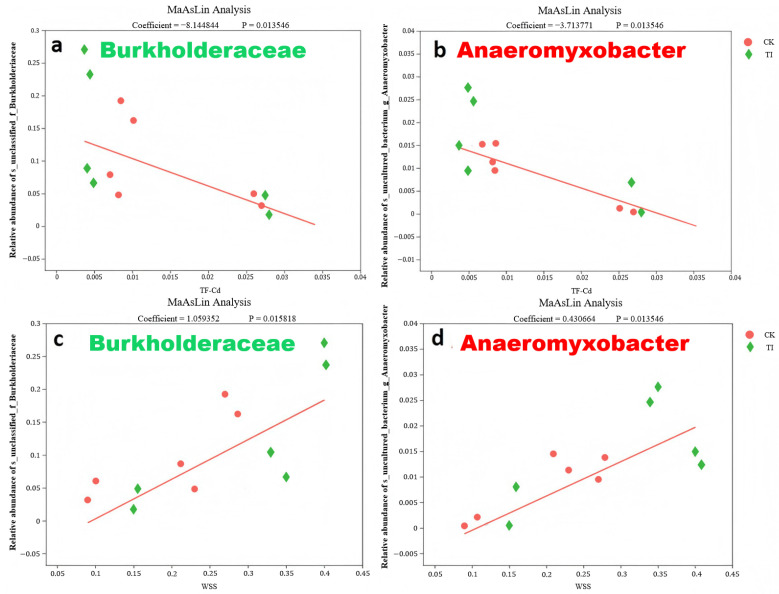
The relation between the relative abundance of *Burkholderiace*, *Bacterium*_g_*Anaeromyxobacter* and Cd translocation factor from root to grain (**a**,**b**), WSS content (**c**,**d**) in rice plant, respectively.

**Table 1 toxics-13-00755-t001:** Physical–chemical properties of soil in paddy field.

Physical-Chemical Indexes	Site A	Site B	Site C
pH	6.3	7.1	7.5
Olsen-P/mg kg^−1^	22.44	34.87	42.25
Olsen-K/mg kg^−1^	62.25	63.54	61.75
Alkaline-N/mg kg^−1^	80.5	78.6	76.8
Total-N/g kg^−1^	2.57	2.56	2.87
Total Cd/mg kg^−1^	1.2	0.85	0.54
Available Cd/mg kg^−1^	0.21	0.15	0.12
Organic Matter/g kg^−1^	45.95	42.52	38.54

**Table 2 toxics-13-00755-t002:** The adsorptive site concentration obtained from stem by the potentiometric titration.

Site	Symbol	Adsorptive Site Concentration (mmol/g) ^A^							
		Total		4 < pKa < 6		pKa ≈ 7		8 < pKa < 11	
Site A	A	4.61 ^a^	0.08	0.96 ^a^	0.02	1.72 ^a^	0.04	1.93 ^b^	0.02
	ATI	6.53 ^c^	0.07	0.91 ^a^	0.02	1.46 ^b^	0.03	4.16 ^a^	0.05
Site B	B	4.90 ^a^	0.06	1.17 ^a^	0.01	1.76 ^a^	0.02	1.97 ^b^	0.04
	BTI	6.80 ^b^	0.08	1.02 ^a^	0.01	1.43 ^b^	0.02	4.35 ^a^	0.08
Site C	C	5.80 ^a^	0.09	0.95 ^a^	0.02	1.82 ^a^	0.03	3.03 ^b^	0.05
	CTI	7.21 ^c^	0.10	0.86 ^a^	0.01	1.81 ^a^	0.03	4.54 ^a^	0.06

^A^ Average apparent acidity constant conditional to I = 0.001 mol L^−1^. Different letters indicate statistically significant differences between treatments by Student’s *t*-test.

**Table 3 toxics-13-00755-t003:** The abundance of the metal-transport-relative enzyme in endophytic bacterial community in rice plant with different treatment.

Enzyme	CK	TI
P-type Ca^2+^ transporter	5774	5921
P-type Zn^2+^ transporter	11,380	13,344
P-type Mg^2+^ transporter	1315	1435
Cd^2+^-exporting ATPase	11,380	13,344
ABC-type Mn^2+^ transporter	115	181
P-type Cu^2+^ transporter	27,352	28,362

## Data Availability

The original contributions presented in this study are included in the article. Further inquiries can be directed to the corresponding author.

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
