# Peer review of "Foliar Transpiration Inhibitor Reduces Cd Accumulation in Rice Grain: The Potential Effect of the Endophytic Bacterial Community"

_toxics, 2025, doi:10.3390/toxics13090755_

Round 1

Reviewer 1 Report

Comments and Suggestions for Authors

Dear authors,

I rate the quality of the article as generally good. It is distinguished by its relevance and scientific novelty.  Judging by the section "Materials and methods", the experiment was carefully and competently planned. A randomized scheme was applied, modern analytical methods were used to quantify cadmium and microbial dna diversity. Most of the conclusions follow from the results obtained. The illustrations are relevant and informative. The local disadvantages of the article are listed below.

  1. When discussing the results of the study, all the observed effects are associated only with endophytic prokaryotes. At the same time, the authors did not study endophytic fungi. The possibility of their contribution to the observed changes is almost not discussed.
  2. The placement of the table 3 and figure 5 in the discussion section seems controversial to me. Because, in my opinion, the correlation matrix represents statistically processed data, i.e. it contains the results.
  3. Describe the statistical analysis in more detail in the materials and methods. Whether the data corresponded to a normal distribution. Which post-hoc tests were used. Which indicator is shown as a bars in the diagrams (standard error or standard deviation). How many repetitions of each analysis were performed.
  4. Table 2. In the note below the table, explain the arrangement of letters to the right of the values. Was the significance of the difference in averages determined within each location or between three locations simultaneously? Why are some columns missing the letter a and the letter arrangement starts with the letter b?
  5. The manuscript does not describe how the data presented in table 3 was obtained. The measurement units are not specified in Table 3. Based on which test is it concluded that the values in the TI column are higher than the values in the CK column? The results of these tests are not presented.
  6. Abbreviations P, N, K, CD, TF are not deciphered in the captions of Figure 5. Is the data in table 1 meant? Which of the two indicators N was used in the calculation? Which of the three TF indicators? What results of cadmium measurement are implied? The original ones? Is it cadmium in the soil at the end of the experiment? Is it cadmium in plants?
  7. The legend of Figures 4b,d,e has a small letter size. The legend and the name of the axes are difficult to read.

Author Response

Comments 1:

When discussing the results of the study, all the observed effects are associated only with endophytic prokaryotes. At the same time, the authors did not study endophytic fungi. The possibility of their contribution to the observed changes is almost not discussed.

Response 1

Your opinion is right. The other endophytic fungi might be contribution to the Cd transportation in paddy plant. However, the contribution should be lower than that of endophytic bacterium. The number of OTUs (Operational Taxonomic Units) (reaching up to 610 in the leaf) and relative abundance of bacteria are generally higher than those of fungi (Wang Pei. Structural characteristics of rice foliar microbial communities and their responses to typical interaction factors such as rice leaf blast [D]. Hunan Agricultural University, 2021.DOI:10.27136/d. cnki.ghunu. 2021. 000231.). And the number of OTUs reached up to 716 after processing by the TI in our experiment. The effect of endophytic fungi should be investigated next step.

Comments 2:

The placement of the table 3 and figure 5 in the discussion section seems controversial to me. Because, in my opinion, the correlation matrix represents statistically processed data, i.e. it contains the results.

Response 2: 

We have transferred the relevant contents in the original Table 3 and 4.2 to Section 3.6. The original Figure 5 and some related contents of 4.3 have been transferred to 3.7, and at the same time, the elements represented by each item on the horizontal axis have been marked in the title.The revised section was present as follow:

3.6. Enhanced Cd Biosequestration Potential in Endophytes

To further illustrate the potential ability of endophytes to biosequester Cd, the abundance of the metal-transport-relative enzyme in endophytes induced by different treatments was shown in table 3. The abundance of P-type Ca2+/Zn2+/Mg2+/Mn2+/Cu2+ transporter and Cd2+-exporting ATPase for endophytes indcuced by TI treatment was higher than that by CK. The Cd2+ can transport into endophytes’ cell through these P-type Ca2+/Zn2+/Mg2+/Mn2+/Cu2+ transporter[23]. Furthermore, the high abundance of Cd2+-exporting ATPase revealed that the endophytes induced by the TI treatment with high Cd-detoxicity ability. It was demonstrated that the endophytes in rice plant reduced by TI were with highest Cd-sequestering ability than that by CK. And this result was corresponding to the endophytes with high stress-tolerant function induced by TI treatment.  

Table 3. the abundance of the metal-transport-relative enzyme in endophytic bacterial community in rice plant with different treatment.

Enzyme

CK

TI

P-type Ca2+ transporter

5774

5921

P-type Zn2+ transporter

11380

13344

P-type Mg2+  transporter

1315

1435

Cd2+ -exporting ATPase

11380

13344

ABC-type Mn2+  transporter

115

181

P-type Cu2+ transporter

27352

28362

 3.7. Identification of typical endophytes and their environmental relevance

In this study, nutrients (N, P, K), WSS and Cd, TFroot-grain in rice stem, were taken into consideration to evaluate the relative contributions to endophytic bacteria on species level. The detailed correlation between the bacteria species and the environmental parameters was shown in Fig.6. A species-level heatmap split the six parameters into two groups at the first level. One was composed of Cd, TFroot-grain and K, N, the other was composed of WSS, P. 

Figure 6. The heatmap of the correlation between relative abundance of endophytic bacterial species and physical parameters. WSS, P, N and K refer to the content of water-soluble saccharide, SP, SN and SK in the rice stem, respectively. And Cd, TF refer to the content of Cd in grain and TFroot-grain, respectively.

Notably, unclassified_g__Mycobacterium was positively correlated with Cd  (R=0.77, 0.001<p≤0.01). metagenome_g__Pseudolabrys was positively correlated with Cd (R=0.89, 0.01<p≤0.05)  and K (R=0.83, 0.01<p≤0.05).  Burkholderiaceae was negatively correlated with Cd (R=-0.94, 0.001<p≤0.01), TFroot-grain (R=-0.88 0.01<p≤0.05), while positively corresponded to WSS (R=0.89, 0.01<p≤0.05).  uncultured_bacterium_g__Anaeromyxobacter was negatively correlated with TFroot-grain (R=-0.86, 0.01<p≤0.05), while positively corresponded to WSS (R=0.86, 0.01<p≤0.05).

The relationship between the WSS content and the microbial species was investigated as well. Two microbes were positively correlated with WSS (p≤0.05), i.e. Burkholderiace and Bacterium_g_Anaeromyxobacter. It was indicated that the relative abundance and the amount of obtained endophytic bacteria increased might lead to decreasing of TFroot-rice. And the detailed MaAslin analysis between two microbes and both the TFroot-rice and WSS content were shown in the Fig. 7.  The high relative abundance of Burkholderiace and Bacterium_g_Anaeromyxobacter was revealed in the rice plant with different treatments, respectively.

Fig.7 The relation between the relative abundance of  Burkholderiace, Bacterium_g_Anaeromyxobacter and Cd translocation factor from root to grain (a, b),  WSS content (c, d) in rice plant, respectively.

Comments 3:

Describe the statistical analysis in more detail in the materials and methods. Whether the data corresponded to a normal distribution. Which post-hoc tests were used. Which indicator is shown as a bars in the diagrams (standard error or standard deviation). How many repetitions of each analysis were performed.

Response 3: 

The statistical analysis has been revised in the materials and methods section. The revised section was present as follow:

2.7. Statistical Analyses

The statistical analysis of the obtained results was used the SPSS 19.0. Every site was set two treatments, with and without spraying with foliar TI. Every treatment included three samples as three replicates. Due to the small sample size, although we conducted a normality test, its effectiveness was insufficient. Considering that this indicator is usually considered to be approximately normally distributed, we used the Student's t-test examine statistical significance of differences in different groups. Mean±SE; standard error was calculated using three replicates for each treatment. 

The correlation between relative abundance of endophytic bacterial species and physical parameters, using Spearman analysis. And the Shapiro-wilk test the data whether corresponding to the normal distribution before the correlation analysis.

Figure 1. Effect of TI on Cd concentrations in rice grains (a); stem nodes (b); roots (c) and Cd translocation factor from root to rice grain (d) of rice plants grown in Cd-contaminated field with different Cd concentration. (The letter a, b indicate statistically significant differences between treatments by Student’s t-test. Mean±SE; standard error was calculated using three replicates for each treatment.).

Figure 2. The translocation factors (TF) of root to stem nodes (a), and (b) stem node to rice grain, following the CK and T1 treatments in different sites. (The letter a, b indicate statistically significant differences between treatments by Student’s t-test. Mean±SE; standard error was calculated using three replicates for each treatment.).

Figure 3. The content of water-soluble saccharide in stem nodes of rice plant (a). The relation between the TFroot-grain and the WSS concentration (b). (The letter a, b indicate statistically significant differences between treatments by Student’s t-test. Mean±SE; standard error was calculated using three replicates for each treatment.)

Table 2. The adsorptive site concentration obtained from stem by the potentiometric titration.

Site

Adsorptive site concentration (mmol/g)*

Symbol 

Total  

4<pKa<6

pKa≈7

8<pKa<11

Site A

A

4.61 b

0.08

0.96 a

0.02

1.72 a

0.04

1.93 b

0.02

ATI

6.53 a

0.07

0.91 a

0.02

1.46 b

0.03

4.16 a

0.05

Site B

B

4.90 b

0.06

1.17 a

0.01

1.76 a

0.02

1.97 b

0.04

BTI

6.80 a

0.08

1.02 a

0.01

1.43 b

0.02

4.35 a

0.08

Site C 

C

5.80 b

0.09

0.95 a

0.02

1.82 a

0.03

3.03 b

0.05

CTI

7.21 a

0.10

0.86 a

0.01

1.81 a

0.03

4.54 a

0.06

*Average apparent acidity constant conditional to I= 0.001 mol L-1.

Different letters indicate statistically significant differences between treatments by Student’s t-test.

Comments 4:

Table 2. In the note below the table, explain the arrangement of letters to the right of the values. Was the significance of the difference in averages determined within each location or between three locations simultaneously? Why are some columns missing the letter a and the letter arrangement starts with the letter b?

Response 4: 

The main focus is on modifying the letters represented by significant differences, and the importance of the average difference is determined within each location. The revised section was present as follow:

Table 2. The adsorptive site concentration obtained from stem by the potentiometric titration.

Site

Adsorptive site concentration (mmol/g)*

Symbol 

Total  

4<pKa<6

pKa≈7

8<pKa<11

Site A

A

4.61 b

0.08

0.96 a

0.02

1.72 a

0.04

1.93 b

0.02

ATI

6.53 a

0.07

0.91 a

0.02

1.46 b

0.03

4.16 a

0.05

Site B

B

4.90 b

0.06

1.17 a

0.01

1.76 a

0.02

1.97 b

0.04

BTI

6.80 a

0.08

1.02 a

0.01

1.43 b

0.02

4.35 a

0.08

Site C 

C

5.80 b

0.09

0.95 a

0.02

1.82 a

0.03

3.03 b

0.05

CTI

7.21 a

0.10

0.86 a

0.01

1.81 a

0.03

4.54 a

0.06

*Average apparent acidity constant conditional to I= 0.001 mol L-1.

Different letters indicate statistically significant differences between treatments by Student’s t-test.

Comments 5:

The manuscript does not describe how the data presented in table 3 was obtained. The measurement units are not specified in Table 3. Based on which test is it concluded that the values in the TI column are higher than the values in the CK column? The results of these tests are not presented.

Response 5: 

experiment section was revised and added the descriptions to the table 3. And the unit was explained. The revised section presents as follow:

2.6. Bioinformatics and statistical analysis

Raw pyrosequencing data were de-multiplexed and quality-filtered, by using Trimmomatic tool in the way Lohse et al.[18] described. Overlapping reads were then merged into single long reads with the FLASH software tool[19]. Qualified sequences were then clustered into OTUs at a 97% similarity cutoff using Usearch v7.1 (http://qiime.org/). The phylogenetic affiliation of each 16S rDNA sequence was analyzed with the RDP Classifier v2.2 (http://sourceforge.net/projects/rdp-classifier/), using a confidence threshold of 0.7 and the reference database Silva (Release 115, http://www.arb-silva.de). The heatmap figures were produced using package 'gplots' in the R (v3.1.1) software (http://www.Rproject.org/).

Gene Abundance Quantification the abundance of each gene in the non-redundant catalog within each sample was quantified by mapping the high-quality, host-filtered reads back to the catalog using Bowtie2 (v2.4.4). The number of reads mapped to each gene was counted. The abundance of a given gene was calculated and normalized as Reads Per Kilobase per Million mapped reads (RPKM) to account for gene length and sequencing depth variations between samples. The formula is:

RPKM = (Number of reads mapped to the gene × 109) / (Total number of mapped reads × Gene length in bases)

This normalized value represents the relative abundance of a specific gene in the microbial community.

Functional Annotation The non-redundant protein sequences were aligned against the Kyoto Encyclopedia of Genes and Genomes (KEGG) database (Release 106.0) using Diamond (v2.0.15) with a blastp search (e-value ≤ 10-5). Genes were assigned to KEGG Orthology (KO) entries based on the best hit.

Identification and Abundance Profiling of Metal-Transport-Related Enzymes KO entries associated with metal transport functions were identified by querying the KEGG BRITE hierarchies and pathway maps (e.g., ko02010 for ABC transporters, ko01100 for Metabolic pathways) for relevant terms.

The relative abundance of a specific metal-transport-related enzyme in a sample is represented by the sum of the RPKM values of all non-redundant genes annotated to the corresponding KO entry. This value serves as a proxy for the potential functional potential of that enzyme within the endophytic bacterial community.

The overall functional profile for metal transport was constructed by aggregating the abundances of all KO entries belonging to this category

Comments 6:

Abbreviations P, N, K, CD, TF are not deciphered in the captions of Figure 5. Is the data in table 1 meant? Which of the two indicators N was used in the calculation? Which of the three TF indicators? What results of cadmium measurement are implied? The original ones? Is it cadmium in the soil at the end of the experiment? Is it cadmium in plants?

Response 6: 

There were some errors in description. After revision, the WSS, P, N and K refer to the content of water-soluble saccharide, SP, SN and SK in the rice stem, respectively. And Cd, TF refer to the content of Cd in grain and TFroot-grain, respectively. Therefore, the revised section in text present as follow:

Figure 6. The heatmap of the correlation between relative abundance of endophytic bacterial species and physical parameters. WSS, P, N and K refer to the content of water-soluble saccharide, SP, SN and SK in the rice stem, respectively. And Cd, TF refer to the content of Cd in grain and TFroot-grain, respectively.(*0.01<p≤0.05, **0.001<p≤0.01)

2.4. Determination and analysis

The nutrition for endophytic microorganism in the rice stem. Weigh 0.5000 g of ground rice stems and add 25 mL of a 2% acetic acid solution at 0.5 mol/L. Shake and extract for 30 minutes at 25 ° C and 200 rpm. Then centrifuge at 10000 × g for 15 minutes, filter the supernatant through a 0.45 μ m membrane, and use the resulting filtrate for measurement. The soluble total nitrogen (SN), Soluble total phosphorus (SP), Soluble potassium (SK) was carried out using potassium persulfate oxidation UV spectrophotometry, molybdenum antimony colorimetric method and inductively coupled plasma optical emission spectroscopy (ICP-OES), respectively (Bao, S.D., 2000. Soil and Agricultural Chemistry Analysis (3rd ed.). China Agriculture Press.).

Comments 7:

The legend of Figures 4b,d,e has a small letter size. The legend and the name of the axes are difficult to read.

Response 7:

To address the issue of overly small fonts in Figures 4b, d, and e, we split Figure 4 into Figure 4 and Figure 5, and enlarged the overly small fonts.Therefore, the revised section in text present as follow:

3.4. Effects of TI on biological species in rice plants.

Figure 4. The Shannon index of different treatments with A, B, C, and ATI, BTI, CTI (a). the abundance of endophytic bacterial community on phylum level in the rice plant with different treatments (b). The community heatmap on species level of different treatments (c). The A, B, C was the CK in siteA, B, C and the ATI, BTI, CTI was the samples in siteA, B, C after treating with TI, respectively. Mean±SE; standard error was calculated using three replicates for each treatment.

3.5. The main predicted function of endophytic bacterial community in different treatments.

Figure 5. The relative abundance of main endophytic bacteria leading to forms-biofilms (a)  and stress-tolerant (b) function in the rice plant with CK and TI treatments.

Reviewer 2 Report

Comments and Suggestions for Authors

Dear,

In the paper the possibility of reducing the cadmium content in plants and rice seeds under the influence of foliar transformation inhibitors was examined.

The topic of the paper is important and the setting of the experiment is appropriate.

However, the presentation of the work is at a very low level, with a lot of technical errors, so it is very difficult to follow. Sentences are incomplete, without punctuation, etc., specially from line 339 onwards. There are also errors in the graphics. For example, Figure 1. There is no letter c which is cited in the Figure legend, etc.

Because of that I consider that the paper should be thoroughly revised and resubmitted.

Best regards,

Author Response

Comments 1:

However, the presentation of the work is at a very low level, with a lot of technical errors, so it is very difficult to follow. Sentences are incomplete, without punctuation, etc., specially from line 339 onwards. There are also errors in the graphics. For example, Figure 1. There is no letter c which is cited in the Figure legend, etc.

Response 1:

Thank you for pointing this out. We have made multiple revisions in response to the questions and suggestions you raised. These revisions include:

1. Some grammatical errors were found throughout the article, and certain optimizations were made to some expressions. This includes the correction of the errors in the original 339 lines in Section 4.1 as raised in the opinions. The revised section was present as follow:

Subsequently, the transportation of Cd was depressed after the foliar application with TI at the heading stage of rice plant growth.

2. We have made corrections to the confusion of information in the chart, including the title of Figure 1. Including supplementing the information in Figure 6. And further explanations were provided for the elements in some of the figures in the article.

2. We supplemented The previously missing experimental methods, including the nutrition for endophytic microorganism in the rice stem in section 2.4. Relevant content, 2.6. Some information in the section of Bioinformatics and statistical analysis. And more detailed modifications were made to the 2.7. Statistical Analyses section. The revised section was present as follow:

2.4. Determination and analysis

The nutrition for endophytic microorganism in the rice stem. Weigh 0.5000 g of ground rice stems and add 25 mL of a 2% acetic acid solution at 0.5 mol/L. Shake and extract for 30 minutes at 25 ° C and 200 rpm. Then centrifuge at 10000 × g for 15 minutes, filter the supernatant through a 0.45 μ m membrane, and use the resulting filtrate for measurement. The soluble nitrogen (SN), soluble phosphorus (SP), soluble potassium (SK) was carried out using potassium persulfate oxidation UV spectrophotometry, molybdenum antimony colorimetric method and inductively coupled plasma optical emission spectroscopy (ICP-OES), respectively[18]

2.6. Bioinformatics and statistical analysis

Gene Abundance Quantification the abundance of each gene in the non-redundant catalog within each sample was quantified by mapping the high-quality, host-filtered reads back to the catalog using Bowtie2 (v2.4.4). The number of reads mapped to each gene was counted. The abundance of a given gene was calculated and normalized as Reads Per Kilobase per Million mapped reads (RPKM) to account for gene length and sequencing depth variations between samples. The formula is:

RPKM = (Number of reads mapped to the gene × 109) / (Total number of mapped reads × Gene length in bases)

This normalized value represents the relative abundance of a specific gene in the microbial community.

Functional Annotation The non-redundant protein sequences were aligned against the Kyoto Encyclopedia of Genes and Genomes (KEGG) database (Release 106.0) using Diamond (v2.0.15) with a blastp search (e-value ≤ 10-5). Genes were assigned to KEGG Orthology (KO) entries based on the best hit.

Identification and Abundance Profiling of Metal-Transport-Related Enzymes KO entries associated with metal transport functions were identified by querying the KEGG BRITE hierarchies and pathway maps (e.g., ko02010 for ABC transporters, ko01100 for Metabolic pathways) for relevant terms.

The relative abundance of a specific metal-transport-related enzyme in a sample is represented by the sum of the RPKM values of all non-redundant genes annotated to the corresponding KO entry. This value serves as a proxy for the potential functional potential of that enzyme within the endophytic bacterial community.

The overall functional profile for metal transport was constructed by aggregating the abundances of all KO entries belonging to this category. 

2.7. Statistical Analyses

The statistical analysis of the obtained results was used the SPSS 19.0. Every site was set two treatments, with and without spraying with foliar TI. Every treatment included three samples as three replicates. Due to the small sample size, although we conducted a normality test, its effectiveness was insufficient. Considering that this indicator is usually considered to be approximately normally distributed, we used the Student's t-test examine statistical significance of differences in different groups. Mean±SE; standard error was calculated using three replicates for each treatment.

The correlation between relative abundance of endophytic bacterial species and physical parameters, using Spearman analysis. And the Shapiro-wilk test the data whether corresponding to the normal distribution before the correlation analysis.

4. We split Figure 4 into Figure 4 and Figure 5. So as to better present the necessary information in the picture to the readers.

5. To further present our experimental results in a clear and focused manner, we have moved Table 3, the original Figure 5, and their related descriptions from the discussion section to the results section.

Reviewer 3 Report

Comments and Suggestions for Authors

The manuscript titled "Foliar transpiration inhibitor reduces Cd accumulation in rice grain: the potential effect of the endophytic bacterial community" presents results of effect of foliar transpiration inhibitors (TI) on the Cd content and endophytic bacteria community in rice plant. The paper is concise, well-written, and provides new viewpoint to understand the reducing of Cd accumulation in the rice grain after the foliar reagents, which is of potential effect on the rice safety-production.

However, there are some minor points that require improvement:

- The authors should provide the full name for the abbreviation CK at the first place where it is mentioned.

- First sentence in Result subsection 3.3. The adsorptive site concentration of stem nodes. 258-260 “To investigate the adsorptive capacity of stem nodes to Cd, the adsorptive site concentration of stem nodes was analyzed using potentiometric titration and the results were shown in Table 2” present methods, so should be or moved to Material & Methods section, or removed.

- In the Discussion section, I recommended that the authors avoid using subsections. The results of the study should be discussed as a whole rather than separately, which will provide a much clearer understanding of the findings and the overall aim of the work.

- Table 3 and Figure 5 should be presented in the Results section.

Author Response

Comments 1:  

The authors should provide the full name for the abbreviation CK at the first place where it is mentioned

Response 1:

Thank you for pointing this out. We marked the full name of CK on line 122 of the article.The revised section was present as follow:

2.3. Experimental design

Beginning at mid-July 2019, TI were sprayed on rice plants firstly, and then sprayed once a week from heading stage to the grouting stage. The total spray number was three in overall rice growth process. These foliar reagents were sprayed the leaves with a handheld sprayer. Control plants (CK) were sprayed with water and every plot with 20 m2 were sprayed with 2.0 L foliar reagents/water. At maturity stage, three plants with roots and topsoil (0-20 cm) were dug out by a shovel from the center of each plot. Wash the roots with water, then bring them back to the lab and wash them three times with deionized water. The harvested plants are then separated into grains, roots, and stems. These grains and stems were dried for 6 hours at 70℃, and the roots were stored in a constant temperature refrigerator at -70 ℃.

Comments 2:

First sentence in Result subsection 3.3. The adsorptive site concentration of stem nodes. 258-260 “To investigate the adsorptive capacity of stem nodes to Cd, the adsorptive site concentration of stem nodes was analyzed using potentiometric titration and the results were shown in Table 2” present methods, so should be or moved to Material & Methods section, or removed.

Response 2: 

The description of potentiometric titration in Subsection 3.3 has been presented in the Materials and Methods section, and we have also chosen to modify the description here. The revised section was present as follow:

2.4. Determination and analysis (Lines 156 to 171)

Adsorptive site concentration measured with potentiometric titration.  The potentiometric titrations were performed with the Metrohm 905 Titrando system, coupled with a GK2401C combination glass electrode. The glass electrode was calibrated by using buffers at pH of 4.01, 7.00 and 10.00 before each titration. The concentration of the standard HCl and NaOH solutions were tested before the titration.

The fresh stem nodes were washed with distilled water and ethanol for three times and then vacuum freezing to constant weight. The 0.50g dried stem nodes were ground to powders using liquid N2 in mortar. The powders were placed in beaker with 70 mL of ultrapure water (18 MΩ·cm, 25 °C) with 0.001 M NaCl, under a N2 atmosphere at 25 °C and then titrated with 0.1 M NaOH and 0.5 M HCl solutions. A known amount of HCl was added at the beginning of the experiment to lower the pH to approximately 2.5. The composites were equilibrated for 40 min and then titrated to pH 10 with NaOH. The background value was titrated with the same treatment using deionized water. Subsequently, a non-electrostatic model was used to fit the potentiometric titration data. The pKa (type of functional group), total adsorption sites, and site concentration were calculated using FITEQL4.

3.3. The adsorptive site concentration of stem nodes.

The results of the cadmium adsorption capacity of stem nodes are shown in Table 2. The total adsorptive site concentration of TI and CK in site C was 6.83 and 5.10 mmol/g, respectively. The total adsorptive site concentration was increased in TI than that of CK, regardless of different sites. This result corresponded to the TI induced lower TFroot-rice of Cd than that of CK, because of high concentration of adsorptive sites for Cd in stem nodes. The functional groups present on biosurfaces have been divided into three types[22], where pKa values of 4-6, ~7, and 8-11 correspond to carboxyl group, phosphate group, and amino/hydroxyl group, respectively. The TI treatment can increase the amino/hydroxyl groups, as shown in Table 2.

Comments 3:

In the Discussion section, I recommended that the authors avoid using subsections. The results of the study should be discussed as a whole rather than separately, which will provide a much clearer understanding of the findings and the overall aim of the work.

Response 3:

Thank you for pointing this out. After our discussion, we believe that we will still choose to present our discussion part in the form of three sections here. The reasons are as follows

1. The method of dividing into sections can clearly distinguish different discussion focuses, and each section revolves around a core theme. This modular structure helps readers gradually understand the research findings.

2. There is a considerable amount of information in the three sections. Retaining the structure of the three sections can effectively prevent the possible confusion of information after merging, thereby guiding readers to naturally transition from one topic to another.

3. Besides, we also think that in the next section, namely the Conclusions section, readers can also get a clear understanding of the findings and the overall aim of the work.

Comments 4:

Table 3 and Figure 5 should be presented in the Results section.

Response 4:

We have transferred the relevant contents in the original Table 3 and 4.2 to Section 3.6. The original Figure 5 and some related contents of 4.3 have been transferred to 3.7, and at the same time, the elements represented by each item on the horizontal axis have been marked in the title.The revised section was present as follow:

3.6. Enhanced Cd Biosequestration Potential in Endophytes

To further illustrate the potential ability of endophytes to biosequester Cd, the abundance of the metal-transport-relative enzyme in endophytes induced by different treatments was shown in table 3. The abundance of P-type Ca2+/Zn2+/Mg2+/Mn2+/Cu2+ transporter and Cd2+-exporting ATPase for endophytes indcuced by TI treatment was higher than that by CK. The Cd2+ can transport into endophytes’ cell through these P-type Ca2+/Zn2+/Mg2+/Mn2+/Cu2+ transporter[23]. Furthermore, the high abundance of Cd2+-exporting ATPase revealed that the endophytes induced by the TI treatment with high Cd-detoxicity ability. It was demonstrated that the endophytes in rice plant reduced by TI were with highest Cd-sequestering ability than that by CK. And this result was corresponding to the endophytes with high stress-tolerant function induced by TI treatment.  

Table 3. the abundance of the metal-transport-relative enzyme in endophytic bacterial community in rice plant with different treatment.

Enzyme

CK

TI

P-type Ca2+ transporter

5774

5921

P-type Zn2+ transporter

11380

13344

P-type Mg2+  transporter

1315

1435

Cd2+ -exporting ATPase

11380

13344

ABC-type Mn2+  transporter

115

181

P-type Cu2+ transporter

27352

28362

 3.7. Identification of typical endophytes and their environmental relevance

In this study, nutrients (N, P, K), WSS and Cd, TFroot-grain in rice stem, were taken into consideration to evaluate the relative contributions to endophytic bacteria on species level. The detailed correlation between the bacteria species and the environmental parameters was shown in Fig.6. A species-level heatmap split the six parameters into two groups at the first level. One was composed of Cd, TFroot-grain and K, N, the other was composed of WSS, P. 

Figure 6. The heatmap of the correlation between relative abundance of endophytic bacterial species and physical parameters. WSS, P, N and K refer to the content of water-soluble saccharide, SP, SN and SK in the rice stem, respectively. And Cd, TF refer to the content of Cd in grain and TFroot-grain, respectively.

Notably, unclassified_g__Mycobacterium was positively correlated with Cd  (R=0.77, 0.001<p≤0.01). metagenome_g__Pseudolabrys was positively correlated with Cd (R=0.89, 0.01<p≤0.05)  and K (R=0.83, 0.01<p≤0.05).  Burkholderiaceae was negatively correlated with Cd (R=-0.94, 0.001<p≤0.01), TFroot-grain (R=-0.88 0.01<p≤0.05), while positively corresponded to WSS (R=0.89, 0.01<p≤0.05).  uncultured_bacterium_g__Anaeromyxobacter was negatively correlated with TFroot-grain (R=-0.86, 0.01<p≤0.05), while positively corresponded to WSS (R=0.86, 0.01<p≤0.05).

The relationship between the WSS content and the microbial species was investigated as well. Two microbes were positively correlated with WSS (p≤0.05), i.e. Burkholderiace and Bacterium_g_Anaeromyxobacter. It was indicated that the relative abundance and the amount of obtained endophytic bacteria increased might lead to decreasing of TFroot-rice. And the detailed MaAslin analysis between two microbes and both the TFroot-rice and WSS content were shown in the Fig. 7.  The high relative abundance of Burkholderiace and Bacterium_g_Anaeromyxobacter was revealed in the rice plant with different treatments, respectively.

Fig.7 The relation between the relative abundance of  Burkholderiace, Bacterium_g_Anaeromyxobacter and Cd translocation factor from root to grain (a, b),  WSS content (c, d) in rice plant, respectively.

Round 2

Reviewer 1 Report

Comments and Suggestions for Authors

Dear authors, thank you for the detailed and comprehensive answers to the questions. I have no complaints about the revised manuscript that would prevent its publication.

Author Response

We would like to express our sincere gratitude to the reviewer for the thorough review and the generous comments provided. The suggestions were highly insightful and have led to significant improvements in the manuscript.

Reviewer 2 Report

Comments and Suggestions for Authors

The following corrections were required:

Lines 144-147 should be in the past tense or passive.

Line 149: correct the text .. were determined…

Line 202 The…

Line 234-235: Correct the sentence grammatically: In every site two treatments were set up, with and without spraying with foliar TI?

Line 335: correct the sentence, it is unclear: To sum up the TI treatment can increase the relative abundance of plant growth-promoting bacteria.

Line 407-413: The capital letters are missing at the beginning of the sentences

Line 418: Correct the sentence

Line 431-432: Correct the sentence, it is unclear: TI treatment induced a higher TFroot-stem and lower , but a lower TF stem-rice, indicating that it enhanced Cd accumulation in the stem nodes.

Author Response

Comments 1:

Lines 144-147 should be in the past tense or passive.

Response 1:

We replace the expression "to measure" with "for the measurement of". The revised section was present as follow:

The sample was then centrifuged at 4000×g for 10 min at 4 °C and the supernatant was collected for the measurement of the WSS content using Bradford method[17].

Comments 2:

Line 149: correct the text .. were determined…

Response 2: 

We change to the passive voice and replace the original "carried out" with "were determined". The revised section was present as follow:

Weigh 0.5000 g of ground rice stems and add 25 mL of a 2% acetic acid solution at 0.5 mol/L. Shake and extract for 30 minutes at 25 ° C and 200 rpm. Then centrifuge at 10000 × g for 15 minutes, filter the supernatant through a 0.45 μ m membrane, and the resulting filtrate was used  for measurement. The soluble nitrogen (SN), soluble phosphorus (SP), soluble potassium (SK) were determined using potassium persulfate oxidation UV spectrophotometry, molybdenum antimony colorimetric method and inductively coupled plasma optical emission spectroscopy (ICP-OES), respectively[18].

Comments 3:

Line 202 The…

Response 3: 

We added "the". The revised section was present as follow:

Raw pyrosequencing data were de-multiplexed and quality-filtered, by using the Trimmomatic tool in the way Lohse et al.[19]

Comments 4:

Line 234-235: Correct the sentence grammatically: In every site two treatments were set up, with and without spraying with foliar TI?

Response 4:

We have corrected the grammar mistakes.The revised section was present as follow:

Every site had two treatments: foliar TI spray treatment and control.

Comments 5:

Line 335: correct the sentence, it is unclear: To sum up the TI treatment can increase the relative abundance of plant growth-promoting bacteria.

Response 5:

We have modified the original phrase "To sum up the TI treatment can increase the relative abundance of plant growth-promoting bacteria." The revised section was present as follow:

TI treatment can regulate the endophytic bacterial community by significantly increasing the relative abundance of plant growth-promoting bacteria and decreasing the abundance of some non-promoting bacteria.

Comments 6:

Line 407-413: The capital letters are missing at the beginning of the sentences.

Response 6:

Correct the error of no capitalized first letter. The revised section was present as follow:

 Metagenome_g__Pseudolabrys was positively correlated with Cd (R=0.89, 0.01<p≤0.05)  and K (R=0.83, 0.01<p≤0.05).  Burkholderiaceae was negatively correlated with Cd (R=-0.94, 0.001<p≤0.01), TFroot-grain (R=-0.88 0.01<p≤0.05), while positively corresponded to WSS (R=0.89, 0.01<p≤0.05).  Uncultured_bacterium_g__Anaeromyxobacter was negatively correlated with TFroot-grain (R=-0.86, 0.01<p≤0.05), while positively corresponded to WSS (R=0.86, 0.01<p≤0.05).

Comments 7:

Line 418: Correct the sentence

Response 7:

We have modified the structure of this sentence. The revised section was present as follow:

Fig. 7 presents the detailed MaAslin analysis of the relationships between the two microbes and both TFroot-rice and WSS content.

Comments 8:

Line 431-432: Correct the sentence, it is unclear: TI treatment induced a higher TFroot-stem and lower , but a lower TF stem-rice, indicating that it enhanced Cd accumulation in the stem nodes.

Response 8:

We have corrected the grammar mistakes. The revised section was present as follow:

TI treatment resulted in a higher TFroot-stem and a lower TFstem-rice, suggesting that Cd accumulation was enhanced in the stem nodes rather than in the rice grain.